# Challenges of Systemic Therapy Investigations for Bone Sarcomas

**DOI:** 10.3390/ijms23073540

**Published:** 2022-03-24

**Authors:** Kenji Nakano

**Affiliations:** Department of Medical Oncology, Cancer Institute Hospital of Japanese Foundation for Cancer Research, 3-8-31 Ariake, Koto, Tokyo 135-8550, Japan; kenji.nakano@jfcr.or.jp; Tel.: +81-3-3520-0111

**Keywords:** bone sarcoma, osteosarcoma, Ewing sarcoma of bone, chondrosarcoma, AYA cancer, pediatric cancer, tyrosine kinase inhibitor

## Abstract

Bone sarcoma is a rare component of malignant solid tumors that accounts for only ~0.2% of malignancies. Bone sarcomas present various histological types, and genomic mutations differ markedly by the histological types. Although there are vast mutations in various bone sarcomas, most of them are non-actionable, and even potential targetable mutations that are actionable targets in other malignancies have not shown the appropriate responses in clinical trials for bone sarcomas. Investigations of new systemic therapy, including molecular targeted therapies for bone sarcomas, have thus not progressed like those for other solid tumors. Another problem is that high rates of pediatric/adolescent and young adult patients have bone sarcomas such as osteosarcoma, and patient recruitment for clinical trials (especially randomized trials) is challenging. For pediatric patients, evaluations of tolerability and appropriate dose modifications of new drugs are needed, as their findings could provide the threshold for investigating new drugs for bone sarcomas. To solve these problems, improvements in registry systems, real world data, and pediatric extrapolation have been attempted. We review the issues regarding targeted drug investigations for bone sarcomas, focusing on the current clinical evidence and efforts to resolve these issues.

## 1. Introduction

Bone sarcoma accounts for only about 0.2% of all solid malignancies; each year, approximately 3600 patients in the U.S., 4000 in Europe, and 800 in Japan are newly diagnosed with bone sarcoma [1,2,3]. Bone sarcoma exhibits many histological variations; the current World Health Organization (WHO) classification includes more than 20 histologies in bone sarcoma [4]. Osteosarcoma is the most frequently observed as primary bone sarcoma, followed by Ewing sarcoma and chondrosarcoma [5]. Bone sarcomas can originate from any bone in the body, and even extraskeletal lesions have been observed as the primary lesion [6].

As treatment strategies, surgical resection has been the mainstay as a curable treatment for bone sarcoma. Since the 1970s, the development of cytotoxic chemotherapy has provided improvements in the prognoses of bone sarcoma patients, especially for those with osteosarcoma or Ewing sarcoma. After the 2000s, however, the prognoses of bone sarcoma patients have remained at a plateau, though many new targeted drugs and immunotherapies have been established for cancer therapy [7].

The problems preventing the development of new molecular targeted drugs for bone sarcoma stem from not only the rarity and variety of bone sarcomas resulting in difficulties in finding appropriate therapeutic targets, but also from challenges such as the high rates of pediatric and young adult patients and the difficulty in evaluating the efficacy of treatments without large-scale randomized clinical trials. These challenges may have kept pharmaceutical developers from conducting aggressive investigations of new drugs for bone sarcomas. In this review, we outline the current status of systemic chemotherapy for bone sarcoma and the challenges for future therapeutic development.

## 2. Osteosarcoma

### 2.1. Overview of the History of Systemic Therapy for Osteosarcoma

Osteosarcoma is the most major bone sarcoma, accounting for 20%–40% of all bone sarcomas [5]. The median patient age at the diagnosis of osteosarcoma is 20 years old, and the peak incidence of osteosarcoma is in adolescence and young adulthood, but osteosarcoma accounts for only ~2–3% of all malignant diseases in these populations [1,8]. Regarding genetic risk factors for osteosarcoma, it has been shown that Li-Fraumeni syndrome, known for the mutation of tumor suppressor gene *TP53*, is associated with osteosarcoma [9].

Due to this cancer’s high risks of recurrence and metastasis, the prognosis of osteosarcoma patients was very poor in the era without systemic chemotherapy; the long-term survival of localized osteosarcoma patients was only ~20% or less [10]. Perioperative chemotherapies with cytotoxic antitumor drugs were investigated in the 1970s, and the long-term survival rate then improved remarkably to 60%–70% [11]. After various single-agent and multidrug combination clinical trials, methotrexate, doxorubicin, cisplatin, and ifosfamide have been considered the key drugs in systemic treatments of osteosarcoma (of note, methotrexate is contraindicated for patients aged ≥40 years because of the toxicity risk). As perioperative therapy, neoadjuvant chemotherapy is strongly recommended; the rate of tumor necrosis provided by neoadjuvant chemotherapy is known to be related to prognoses [12,13]. Thus, the current standard perioperative chemotherapy for osteosarcoma consists of MAP (methotrexate, doxorubicin, cisplatin) before and after surgery, and the appropriate indication and timing for adding ifosfamide are under investigation [14,15].

However, if recurrence or metastasis occurs, the prognosis of osteosarcoma patients remains poor; the 5-year overall survival (OS) rate of patients with recurrent or metastatic osteosarcoma has been unchanged at <20% since the 1980s [16,17]. Most long-term survivors of metastatic osteosarcoma have been patients who had limited pulmonary metastatic lesions and underwent a successful metastasectomy [18,19,20], and patients with bone lesions at recurrence have been suggested to have poor OS [21]. Clinical evidence of standard salvage chemotherapy that improves the overall survival of recurrent/metastatic osteosarcoma patients has not been obtained.

### 2.2. Clinical Trials of Therapies Targeting Osteosarcomas

Until ~2005, prospective clinical trials of new systemic therapy for recurrent and/or metastatic osteosarcoma evaluated mainly cytotoxic drugs that were similar to traditional standard regimens [22]. Of them, the addition of muramyl tripeptide (MTP, a synthetic derivative of Bacille Calmette-Guérin) to the standard MAP perioperative chemotherapy was suggested to prolong patients’ survival in the INT-0133 (CCG-7921/POG-9351) randomized clinical trial though the analysis showing that the survival benefit of MTP has some limitations [23], which resulted in the drug’s approval in Europe and some other countries. Topotecan-based chemotherapy and gemcitabine-based chemotherapy have also shown some prospective and/or retrospective efficacy data, but the survival benefits of these regimens are not established and the drugs are not officially approved for the treatment of osteosarcoma [24,25,26,27].

In 2008, the number of phase II trials of molecular targeted drugs and immunotherapy for osteosarcoma exceeded (for the first time) those of cytotoxic chemotherapies, and since then, phase II trials of molecular targeted drugs and immunotherapy have been consistently performed more frequently than those of cytotoxic chemotherapies [28].

A large number of tyrosine kinase inhibitors (TKIs) have been investigated in the 21st century for many malignant diseases, both hematologic diseases and solid tumors, including osteosarcoma. There have been many prospective phase II trials of TKIs as treatments for osteosarcoma, such as sorafenib with or without everolimus, apatinib, regorafenib, cabozantinib and lenvatinib [28,29,30,31,32,33,34,35], mainly targeting vascular endothelial growth factor receptor (VEGFR) and its downstream signaling pathway (Table 1).

In the recent National Comprehensive Cancer Network (NCCN) guideline, some of these TKIs with efficacy data obtained in prospective trials are listed as a treatment option for salvage systemic therapy for recurrent or metastatic osteosarcoma despite this being an off-label use [37].

Monoclonal antibodies are another potential molecular targeted therapy for osteosarcoma: GD2, GPNMB, HER2, and PD-1 are candidate targets of antibody treatment and prospective clinical trials have performed, but none of the monoclonal antibodies has exhibited potential clinical benefits similar to those provided by TKIs.

The disialoganglioside GD2 is a glycosphingolipid that is expressed in some pediatric malignant tumor cells including osteosarcoma [38], and it is thus considered the therapeutic target for an antibody. The investigations of the anti-GD2 antibody dinutuximab were begun in the 20th century, and the results of a phase I trial for pediatric malignancies (neuroblastoma and osteosarcoma) were published in 1998 [39]. Based on the results of a phase III trial (ANBL0032) [40], dinutuximab was approved for pediatric high-risk neuroblastoma as a maintenance therapy, but a phase II trial (AOST1421) of dinutuximab for osteosarcoma did not meet the primary endpoint [41].

GPNMB (glycoprotein non-metastatic b) is a transmembrane glycoprotein normally expressed in cells related to tissue repair, and it is overexpressed in some malignant cells including osteosarcoma [42]. Preclinical studies of glembatumumab vedotin (CDX-011), an antibody-drug conjugate with the combination of anti-GPNMB antibody and monomethyl auristatin E (vedotin), suggested an antitumor effect [43,44], but in the prospective phase II trial (AOST1521), osteosarcoma patients achieved only limited responses to glembatumumab vedotin (of the 22 patients enrolled, an objective response was observed in only one patient) [45].

Human epidermal growth factor receptor 2 (HER2) is well known as the target in many cancer treatments. HER2-targeted therapy is established as the standard treatment for breast cancer and gastric adenocarcinoma [46,47], and it is recently being investigated for other solid tumors such as non-small cell lung cancer and colorectal cancer [48,49]. Overexpression of HER2 is observed in approximately 30% of osteosarcoma cases and was suggested to be poor-prognosis factor [50,51], and thus HER2 could be a candidate for the targeted therapy of osteosarcoma. A phase II trial of the anti-HER2 antibody trastuzumab for osteosarcoma with the combination of conventional chemotherapy did not show clinical benefits [52]. After the failure of trastuzumab, the development of HER2-targeted therapy for osteosarcoma stopped, although new HER2-targeted drugs (antibody, antibody-drug conjugates or TKIs) were investigated and approved for other malignancies.

Immune checkpoint inhibitors were introduced to cancer therapy in the 2010s, and their use has increasingly changed the standard therapy for many malignant diseases [53]. Programmed cell death-1 (PD-1) is the representative target of immune checkpoint inhibitors, and programmed cell death-ligand 1 (PD-L1) expression is known as the response biomarker of anti-PD-1 antibody [54]. PD-L1 expression was observed in 14.0%–80.6% of osteosarcoma cells and was suggested to be associated with poor prognosis [55]. However, prospective clinical trials of anti-PD-1 antibody for treating osteosarcoma did not reveal meaningful responses [56,57]. Regarding immunotherapy, the clinical benefit of adding interferon-alpha (IFN-α) to standard perioperative chemotherapy was evaluated in a large-scale randomized clinical trial (EURAMOS-1), and no survival benefit was observed [58].

Some of these potential antibody targets might be candidates for chimeric antigen receptor-modified T-cell immunotherapy; some clinical and preclinical trials evaluated HER2- or GD2-specific chimeric antigen receptor-modified T cell immunotherapy for sarcoma (dominantly osteosarcoma), and the results demonstrated the therapy’s tolerance and safety; the objective response rate has not been reported at this time [59,60].

## 3. Ewing Sarcoma of Bone

### 3.1. Overview of the History of Systemic Therapy for Ewing Sarcoma

Ewing sarcoma is pathologically characterized by small round cell and translocation (dominantly *EWSR1-FLI1*) fusion gene from t(11;22)(q24;q12) translocation, and it is known to be sensitive to cytotoxic chemotherapy. Combination chemotherapy regimens have been investigated and have progressed since the 1970s [61].

The INT-0091 (CCG-7881/POG-8850) trial conducted in 2003 with non-metastatic Ewing sarcoma patients showed that the alternating chemotherapy using vincristine, doxorubicin and cyclophosphamide (VDC; doxorubicin was changed to dactinomycin after the achievement of a cumulative doxorubicin dose at the threshold with the high risk of cardiotoxicity), and ifosfamide and etoposide (IE), was superior to VDC-alone based on the event-free survival (EFS) rate, and the alternating regimen became the standard strategy for Ewing’s sarcoma [62]. Several clinical trials comparing a higher dose intensity or dose density of these alternating regimens were then performed, and a higher dose intensity (i.e., biweekly chemotherapy administration rather than triweekly) showed more effective results [63,64].

High-dose chemotherapy (HDCT) and autologous stem cell transplantation (ASCT) were investigated for Ewing sarcoma patients who were high-risk and refractory to conventional chemotherapy or in a recurrent/metastatic setting [65,66]. According to the recent Euro-E.W.I.N.G. 99 and Ewing-2008 trials, HDCT/ASCT resulted in survival rates that were superior to those provided by salvage chemotherapies for non-metastatic patients with refractory Ewing sarcoma to induction chemotherapy [67]. However, the survival benefit of HDCT/ASCT was not observed for Ewing sarcoma patients with pulmonary and/or pleural metastasis compared to the patients treated with salvage chemotherapy and whole lung irradiation, in terms of OS [68].

The progress made regarding multimodal strategies including optimal combination chemotherapy has resulted in a 5-year survival rate >70% among non-metastatic Ewing sarcoma patients, but for recurrent and/or metastatic patients, the 5-year survival rate remains at <30% [69]. As noted above, neither an alternating regimen nor HDCT/ASCT provided meaningful survival benefits to metastatic patients, and a standard salvage chemotherapy has not yet been established.

### 3.2. Clinical Trials of Targeted Therapy for Ewing Sarcomas

Cytotoxic chemotherapy regimens for Ewing sarcoma were actively investigated until recently. Temozolomide-based salvage chemotherapy was evaluated prospectively for approximately 20 years and showed promising objective responses [70,71,72,73,74], and a temozolomide-based regimen was under investigation as a first-line therapy [75]. Topotecan-based therapies also provided objective clinical responses [24,76,77], but the use of topotecan-based combination chemotherapy as a first-line treatment failed to show survival benefit in a phase III trial [78].

There have been fewer prospective clinical trials of targeted therapy such as TKIs or monoclonal antibodies for Ewing sarcoma compared to osteosarcoma. In a phase II trial of the TKI cabozantinib (which was also mentioned above in the Osteosarcoma section), a cohort of Ewing sarcoma patients was examined and the efficacy and safety of cabozantinib for Ewing sarcoma were evaluated. An objective response was observed in 10 of 39 patients with measurable lesions (objective response rate [ORR]: 26%, 95% confidence interval [CI]: 13–42) [34], and the NCCN guideline describes this TKI as a treatment option for recurrent/metastatic Ewing sarcoma, without official approval [37]. The *EWSR1-FLI1* fusion gene is thought to function as a regulator of the expression of insulin-like growth factor 1 (IGF1), and overexpression of IGF1 receptor (IGF1R) is observed in Ewing sarcoma cells [79]; the IGF1R pathway could thus be a target in Ewing sarcoma cases. Clinical trials of IGF1R inhibitors, such as figitumumab, cixutumumab, and ganitumab, showed limited clinical benefits [80,81,82,83,84]. There has been recent progress in the development of molecular targeted drugs that directly target *EWS-FLI1* [85], but information on their efficacy remains to be obtained.

## 4. Chondrosarcoma

Chondrosarcoma is the third major bone sarcoma, and in adults, chondrosarcoma is the most common primary bone cancer, accounting for approximately 20% of the cases of adult bone sarcoma [5]. Unlike osteosarcoma and Ewing sarcoma, chondrosarcoma is resistant to systemic therapies, at least those using cytotoxic chemotherapeutic agents. Based on an analysis of the Surveillance, Epidemiology, and End Results (SEER) database, systemic chemotherapy did not help prolong the overall survival of dedifferentiated chondrosarcoma patients [86]. There have been reports that adjuvant chemotherapy can improve prognosis, but the evidence is still limited and standard regimens have not been established [87]. Targeted drugs, such as TKIs, might not show benefits for chondrosarcoma. For example, a result of a randomized phase II trial evaluating regorafenib compared to a placebo did not meet the primary endpoint concerning the progression-free survival (PFS) rate at 12 weeks, although the PFS of regorafenib tended to be longer than that of placebo (median PFS: 19.9 weeks on regorafenib and 8.0 weeks on placebo) [88]. In the NCCN guidelines, dasatinib and pazopanib are introduced as treatment options based on non-randomized phase II trials [89,90], but the efficacy data of these TKIs are not very different from those of regorafenib.

A potential targetable mutation of chondrosarcoma, that is, *IDH* gene encoding isocitrate dehydrogenase (IDH), has various isoforms including IDH1 and IDH2 and key metabolic enzymes that convert isocitrate to α-ketoglutarate [91]. *IDH* mutations are observed in more than half of all cases of chondrosarcoma, and most of them are IDH1-related [92]. A clinical trial evaluating the treatment of chondrosarcoma with ivosidenib (an oral inhibitor of mutant IDH1) was performed, and its tolerability was confirmed [93]. Further evaluation of its efficacy is needed, and a phase II clinical trial is ongoing [94].

## 5. Other Bone Sarcomas

The bone sarcomas discussed above (osteosarcoma, Ewing sarcoma of bone, and chondrosarcoma) account for approximately 80% of all bone sarcomas, and the remaining types of bone sarcoma are various but are observed extremely rarely in clinical practice, even at tertiary high-volume centers [95]. Clinical trials for these rare bone sarcomas have rarely been performed with the exception of chordoma, for which some clinical trials of TKIs were reported [89,96,97,98,99,100].

The Connective Tissue Oncology Society (CTOS) recently published a consensus paper in which ‘ultra-rare sarcomas’ were defined and the future development of treatment strategies was proposed; bone sarcomas other than osteosarcoma, Ewing sarcoma and chondrosarcoma were included as ultra-rare sarcomas [101]. It is hoped that this proposal will pave the way for prospective clinical trials to be conducted for ultra-rare sarcomas, toward the eventual regulatory approval of effective treatments.

## 6. Regulatory Problems and Solutions for Investigating New Treatment Strategies for Bone Sarcomas

### 6.1. Challenges of Investigating Bone Sarcomas

As noted above, molecular targeted therapy for bone sarcoma still faces a very long and difficult path. The main reason for the difficulty is the rarity of bone sarcomas and the diversity of pathological diagnoses, but there are also many other factors that prevent the development of new therapies for bone sarcoma. In this section, we will list such problems and discuss the solutions that are currently available.

### 6.2. Identifying the Appropriate Target: Precision Medicine

With the spread of comprehensive gene mutation searches, whole-genome sequence information for bone sarcoma has been reported. However, the genetic variation in bone sarcoma is diverse, and no driver gene that is specific to the histological type has been identified [102,103,104]. Genetic mutations that are effective therapeutic targets in other cancers might not be sufficiently effective against bone sarcoma (e.g., HER2) [52]. It is necessary to develop unique therapeutic targets for bone sarcoma and the appropriate and effective drugs. In the past, detection by individual immunostaining or fluorescence in situ hybridization (FISH) was necessary for each mutation, and the information obtained from a single specimen was limited. However, in recent years, next generation sequence (NGS) has made it possible to detect a large number of mutations simultaneously [105,106]. This makes it possible to detect rare or agnostic mutations. One example is the neurotrophic tyrosine receptor kinase (NTRK) fusion gene, which is an established target of tumor agnostic therapy including bone and soft tissue sarcoma [107].

### 6.3. Utilizing Real-World Data

To date, the highest level of evidence for assessing the efficacy of treatments, not only for cancer, has been randomized controlled trials. In fact, of the TKIs that have efficacy data of prospective clinical trials for osteosarcoma, regorafenib, which is small in size but has shown results in randomized controlled trials, is listed in Category 1 in the NCCN guidelines [36]. Although randomized controlled trials remain important, the small number of cases makes it difficult to conduct large-scale randomized controlled trials.

In rare diseases, it is not uncommon to have to evaluate efficacy based on single-arm response rates, but in the case of bone sarcoma, it is not easy to find patients for whom response rates can be evaluated. One of the reasons for this difficulty is the criteria used for the evaluation of efficacy. In solid tumors, the objective response is usually evaluated based on the RECIST (response evaluation criteria in solid tumors) criteria [108,109], and clinical trials of antitumor drugs including the objective response as an endpoint require that the patients have measurable lesions. With the RECIST criteria, however, bone lesions are usually regarded as non-measurable lesions, and patients with bone sarcoma may thus not be able to participate in clinical trials that evaluate efficacy based on the objective response without lymph node or distant metastatic disease. Although lung metastases are highly observed, as described in Section 2, lung metastatic lesions could be the target of salvage surgery for bone sarcoma patients if their sizes are measurable and their numbers are limited [110], so the percentage of “not surgically resectable” bone sarcoma patients receiving salvage chemotherapy and requiring new drugs with lung metastases of measurable size is even lower. Bone sarcomas with bone metastases are known to have a poor prognosis [21], and despite the high demand for clinical trials, it is undesirable that bone lesions are not adequately evaluated in clinical trials. A time-to-event outcome with a control group is required when evaluating the efficacy of a treatment for bone sarcoma.

In recent years, the use of real-world data in evaluations of cancer treatment has been attracting attention, and the possibility of using such data as a control group for rare diseases such as bone sarcoma is being investigated [111]. Since off-label antineoplastic drugs are often used for bone sarcoma [112], it is necessary to examine what type of population would be appropriate as a treatment control group.

### 6.4. The Development of Clinical Trials for Child and AYA Patients

As we have discussed, osteosarcoma and Ewing, which account for a high proportion of bone sarcomas, have a high proportion of pediatric and adolescent and young adult (AYA) patients. Younger patients generally have fewer complications and tolerate cytotoxic anticancer agents better, which allows them to receive higher doses of chemotherapy, but their long-term prognosis is still not good enough [113,114,115,116].

While it is important to promote clinical trials for younger patients, there are several challenges to be addressed. For example, the rate of inclusion of AYA cancer patients in clinical trials is lower than the actual incidence of these patients [117,118]. This may be due to the small number of patients who meet eligibility criteria, a lack of awareness of clinical trials by their physicians, and patient education and financial problems [119]. It is thus important to improve the social support of the base in order to increase the participation rate of AYA patients in clinical trials [120].

For pediatric patients, another factor that makes the rapid evaluation of clinical trials difficult is the fact that dose-finding phase I studies must be conducted again from the beginning for new pediatric antitumor drugs, even for drugs that have been approved for efficacy and safety in adults. In recent years, a review of the dosing steps for pediatric patients and the individualization of drug dosages regardless of whether a pediatric population is involved have been studied, and it is expected that the results of efficacy evaluations will be obtained more quickly in the future [121,122,123].

### 6.5. Improvement of Regulatory System

The review system for evaluating and approving new drugs for rare diseases, including bone sarcoma, also needs to be examined. Rare diseases take longer to incorporate into patient populations, and attempting to perform clinical trials on the same scale as those for major diseases would take an enormous amount of time and place a heavy burden on the development companies. It is necessary to examine the evidence required for regulatory approval separately based on the frequency and severity of the disease under consideration, as in the case of the ultra-rare sarcoma described above in Section 5. New review systems, such as those for breakthrough therapy in the U.S. and PRIME in the EU, are being developed as review and approval systems for innovative treatments for rare diseases [124]. The development of tumor-agnostic therapies is also underway, and clinical trials are being designed for therapeutic targets that are expected to be effective regardless of the carcinoma, such as the NTRK targeted therapy described above [107,125]. With these agnostic therapies, it is hoped and expected that a new treatment for bone sarcoma will be developed [126].

## Figures and Tables

**Table 1 ijms-23-03540-t001:** Prospective clinical trials evaluating efficacy of tyrosine kinase inhibitors (TKIs) for osteosarcoma.

Drug/Regimen	Phase	n	Efficacy	Ref.
Sorafenib	II	35	ORR: 8%. PFS at 4 mos.: 46% (95%CI: 28–63) Median PFS: 4 mos. (95%CI: 2–5) Median OS: 7 mos. (95%CI: 7–8)	[29]
Sorafenib + everolimus	II	38	ORR: 5%. PFS at 6 mos.: 45% (95%CI: 28–61) Median PFS: 5 mos. (95%CI: 2–7) Median OS: 11 mos. (95%CI: 8–15)	[30]
Apatinib	II	37	ORR: 43.24%. PFS at 4 mos.: 56.76% (95%CI: 39.43–70.84) Median PFS: 4.50 mos. (95%CI: 3.47–6.27) Median OS: 9.87 mos. (95%CI: 7.97–18.93)	[31]
Regorafenib vs. placebo	II	26 vs. 12	ORR: 8% vs. 0%. PFS at 8 wks: 65% vs. 0% Median PFS: 16.4 wks (95%CI: 18.0–27.3) vs. 4.1 wks (95%CI:3.0–5.7) Median OS: 11.3 mos. (95%CI: 5.9–23.9) vs. 5.9 mos. (1.3–16.4) *	[32]
	II	22 vs. 20	Median PFS: 3.6 mos. (95%CI: 2.0–7.6) vs. 1.7 mos. (95%CI: 1.2–1.8), HR: 0.42 (95%CI: 0.21–0.85)	[33]
Cabozantinib	II	42	ORR: 12% (95%CI: 4–36) PFS at 4 mos.: 71% (95%CI: 55–83) PFS at 6 mos.: 33% (95%CI: 20–50) Median PFS: 6.7 mos. (95%CI: 5.4–7.9) Median OS: 10.6 mos. (95%CI: 7.4–12.5)	[34]
Lenvatinib	I/II	31	ORR: 6.7% (95%CI: 0.8–22.1) PFS at 4 mos.: 29% (95%CI: 14.2–48.0) Median PFS: 3.0 mos.	[35]
Lenvatinib + chemo ^†^	I/II	35 ^‡^	ORR: 9% PFS at 4 mos.: 51% (95%CI: 34–69) Median PFS: 8.7 mos. (95%CI: 4.5–12.0) Median OS: 16.3 mos. (95%CI: 12.6–not estimable)	[36]

* Ten of 12 patients assigned to the placebo group crossed over to open-label regorafenib after progression. ^†^ Ifosfamide plus etoposide. ^‡^ Fifteen patients from phase I and 20 patients from phase II were pooled and evaluated for efficacy. mos.: months, ORR: objective response rate, OS: overall survival, PFS: progression-free survival, wks: weeks.

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
