# Peer review of "Challenges of Systemic Therapy Investigations for Bone Sarcomas"

_ijms, 2022, doi:10.3390/ijms23073540_

Round 1

Reviewer 1 Report

In this review, author took a unique vision to practically generalize current challenges of targeted therapy investigations for bone sarcomas.  Author overviewed the history of systemic therapy for three major bone sarcomas including osteosarcoma, Ewing sarcoma and chondrosarcoma, and then summarized results and status of the most recent clinical trials of therapies targeting bone sarcomas. Author also discussed current regulatory problems and solutions for investigating new treatment strategies for bone sarcomas. The merit to publish the most updated information and studies in manuscript is high. I only have following minor points, which need to be addressed before publishing it.  

  1. Page 1, 2nd paragraph, the statement “Since the 1980s…” is not accurate in this location. Although the citation ref 7 has used data from 1980s to 2000s, it is well-documented that the development of cytotoxic chemotherapy has started from 1970s. Author should revise this statement.
  2. Authors should use full names rather than abbreviations in their manuscripts. For example, in page 6, “approx.”.

Author Response

  1. As you pointed, the statement “Since the 1980s....” is not accurate and I revised it to “1970s...”,
  2. Under your advice, I changed abbreviations in the manuscripts to full names.

Reviewer 2 Report

The author of this Review focuses on the challenges regarding targeted therapy as a treatment for most common bone sarcomas, mainly osteosarcoma, Ewing sarcoma and chondrosarcoma. The author describes main systemic treatments currently available for these diseases and main issues in identifying new therapeutic strategies.

The topic is of interest, since bone sarcomas most frequently occur in pediatric/adolescent patients and progresses in treatment options for these lesions have been very limited.

Overall, the manuscript is very well written and organised, however a little more in depth acknowledgement of preclinical state-of-art in new research perspectives would add value to the manuscript.

For example, a brief description of main genomic rearrangements reported in bone sarcomas, given that the Review focuses on targeted therapy, would improve readability for readers which are naive to the specific pathology. 

In addition, the author should stress the role of NGS analysis for the identification of druggable targets, in this regard recent works have highlighted this topic in bone and soft tissue sarcoma.  

The following references should be added to the manuscript:

  - Next-Generation Sequencing Approaches for the Identification of Pathognomonic Fusion Transcripts in Sarcomas: The Experience of the Italian ACC Sarcoma Working Group. Front Oncol. 2020 Apr 15;10:489. doi: 10.3389/fonc.2020.00489. Erratum in: Front Oncol. 2020 Jun 23;10:944. PMID: 32351889; PMCID: PMC7175964.  

- The clinical utility of next-generation sequencing for bone and soft tissue sarcoma. Acta Oncol. 2021 Oct 22:1-7. doi: 10.1080/0284186X.2021.1992009. Epub ahead of print. PMID: 34686105.

Moreover, the following relevant reference should be added:

Lavit, Elise et al. “Treatment of 120 adult osteosarcoma patients with metachronous and synchronous metastases: A retrospective series of the French Sarcoma Group.” International journal of cancer vol. 150,4 (2022): 645-653. doi:10.1002/ijc.33823

Author Response

  1. Under your advice, I added the description of NTRK rearrangement in the section 6.2, as the example of targetable mutations.
  2. The benefit of NGS was added in the section 6.2.
  3. I added 3 references you suggested in the manuscript.

Reviewer 3 Report

The study aimed to review and discuss available data on the development of new targeted therapies in bone sarcomas.

The topic could be interesting; however, my impression is negative.

The manuscript mostly focuses on available evidence on TT in bone sarcomas.

The part regarding the main objective (challenges) is poorly described. To be honest, most of the challenges are listed in... the introduction (42-47).  The most important (based on the title and aim) is part 6 but it doesn't provide any interesting or novel data/interpretations. The majority of described "challenges" are simply obvious (once again - as described in the introduction).

Moreover, there are several inconsistencies within the manuscript. Examples:

  • "In this review, we outline the current status of systemic chemotherapy for bone sarcoma [...]" --> systemic therapy (with chemotherapy) or only targeted therapy?
  • "With the RECIST criteria, however, bone lesions are usually regarded as non-measurable lesions, and patients with bone sarcoma  may thus not be able to participate in clinical trials that evaluate efficacy based on the  objective response without lymph node or distant metastatic disease" --> but the main indication for targeted therapy in bone sarcomas is METASTATIC disease, not a primary bone lesion, and the most frequent site of metastases are lungs (usually measurable), so I can't see any correlation there,
  • "The development of clinical trials for child and AYA patients"' - I agree with the given arguments, but why was it used particularly in the context of targeted therapies? It's a common problem in whole pediatric medicine.

Author Response

  1. Under your advice, I added some description in the section 6.
  2. Although the manuscript focuses "Targeted Therapy," I believe it is necessary to mention cytotoxic anticancer agents in the text because cytotoxic anticancer agents were more frequently tested in clinical trials for bone sarcoma, especially for osteosarcoma, until 2005, and cytotoxic anticancer agents are still actively being tested in clinical trials in recent years.
  3. As you pointed, the most common site of metastasis in recurrent cases of bone sarcoma is the lung, but there are also many cases with bone lesions, and cases with bone metastasis have a poor prognosis, so there is a high demand for clinical trials. However, there are many cases with bone lesions, and those with bone metastases have a poor prognosis. We have added a note to that effect in Section 6.3.
  4. Regarding the issue of pediatric/AYA generation, since the dosage setting in Phase 1 is an issue for pediatric patients, it was clearly stated that the paragraph is an issue for pediatric cases.

Round 2

Reviewer 3 Report

I am sorry but I still cannot make a positive recommendation.

It is an article about "challenges of targeted therapy investigations", not targeted therapy and systemic therapy in bone sarcomas in general.

As I stated before the aim of the study does not correspond with the content.  "In this review, we outline the current status of systemic chemotherapy for bone sarcoma and the challenges for future therapeutic development." --> but please read the title again. Moreover, I cannot find a link between conventional cytotoxic chemotherapy, immunotherapy (you also mentioned it) and TT. I agree that chemotherapy is (even more) important in Ewing sarcoma than TT but this work should focus on challenges in TT research. Only that and no more. Only paragraph 6 is about the challenges.

"As you pointed, the most common site of metastasis in recurrent cases of bone sarcoma is the lung, but there are also many cases with bone lesions, and cases with bone metastasis have a poor prognosis, so there is a high demand for clinical trials. " --> I do not agree. You wrote "many cases" --> please provide me a reliable reference to convince me. Any statistics? Epidemiological data? The most important problems are lung metastases but they are usually measurable.

Author Response

Under your advice, I changed the title of the article to “Challenges of systemic therapy investigations for bone sarcoma.”

In the section 6.3, as you stated, most recurrent/metastatic bone sarcoma patients have lung metastases, which might be the measurable target lesion by RECIST criteria; however, as I discussed in the section 2, metasectomy of lung metastases is the standard treatment strategies in bone sarcomas, so the percentage of “not surgically resectable” bone sarcoma patients receiving salvage chemotherapy and requiring new drugs with lung metastases of measurable size is even lower. I added these discussions in the section 6.3.